# Using Systems Thinking to Improve Tourism and Hospitality Research Quality and Relevance: A Critical Review and Conceptual Analysis

**Gianna Moscardo** 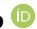

College of Business, Law and Governance, James Cook University, Townsville, QLD 4811, Australia; gianna.moscardo@jcu.edu.au

**Abstract:** This paper argues that that much published tourism and hospitality research has had little influence on tourism or hospitality practice especially with regard to the problems of sustainability because of a failure to use systems thinking to guide research questions and approaches. This critical review and conceptual paper demonstrates how a systems thinking approach could be used to improve both the relevance of, and theoretical development in, tourism and hospitality research in the area of sustainability. This paper reviewed recent published research into tourism's social impacts to demonstrate the power of taking a systems approach to map out the research problem area. It then critically reviewed the use of concepts from psychology in published research into guest engagement in sustainability programs in hospitality businesses to demonstrate the value of systems thinking for organising theoretical concepts. In both of the reviewed areas the overwhelming conclusion was that the majority of the research lacked both practical relevance and was based on inappropriate or deficient theoretical understanding.

**Keywords:** systems thinking; tourism social impacts; guest engagement; hospitality sustainability

## 1. Introduction

Tourism and hospitality, as an identified research area, is studied by researchers from a range of established disciplines. Those who identify themselves as primarily tourism and/or hospitality researchers, rather than as sociologists or psychologists who study tourism and hospitality, are ultimately focused on understanding and solving tourism problems [1]. Tourism and hospitality problems are more than simply the management and marketing issues faced by private sector businesses. Governments, NGOs, communities, tourists and guests, destination residents and staff all face problems and this paper argues that the study of tourism and hospitality is essentially applied as it seeks to address these complex problems. Unfortunately, the evidence suggests that much published tourism and hospitality research has had little influence on tourism or hospitality at any level, especially with regard to the problems of sustainability [2,3].

There has been very little discussion in tourism and hospitality about of why this gap between research and practice exists and how it might be closed or bridged. As with many other issues in tourism there is ample analysis and evidence available from the wider social science literature to shed light on this gap. One major approach to this gap that has been widely used in various areas of management and business [4] is Anderson, Herriot and Hodgkinson's (2001) four fold typology of academic research [5]. This descriptive framework generates four categories based on two main dimensions of methodological rigour and practical relevance. The four categories are:

- Pragmatic research which has both high rigour and high relevance and is the goal of research in applied fields;

- Popularist research which has low rigour but high relevance, often emerging in response to new trends or unexpected changes in areas of interest such as the 2020 COVID 19 pandemic;
- Pedantic research which has high rigour and low relevance with a focus on developing more exact and precise, although not necessarily valid, measurement instruments; and
- Puerile research which has neither relevance nor rigour.

Anderson and colleagues (2001) [5] noted that research outside the pragmatic category is not only of little value in addressing applied problems it can also lead to the use of irrelevant theory and limit the development of theoretical approaches in general. Anderson et al. (2001) [5] also argued that increasing pressure on academics to publish often and quickly, especially in specific journals, and to generate external, often private sector, research funding has pushed them away from pragmatic research towards the other three categories. This is an argument that has continued to gain ground in more critical analyses of the state of academic publishing both within tourism and hospitality [6] and beyond [7–9]. The push for both publication and the appearance of relevance to meet various political pressures on academia has resulted in many academics claiming relevance without any clear description of what they mean by relevance [10,11]. One answer proposed to this issue of relevance and moving towards more pragmatic research lies in the use of systems thinking [4,12–14].

This paper argues that one of the reasons that a research application gap exists in tourism and hospitality, especially with regard to solving the problems of sustainability is that much tourism and hospitality research exists outside the pragmatic category. It further argues that this problem reflects a lack of a sound understanding of the phenomena under study as systems [15]. This critical review and conceptual paper will demonstrate how a systems thinking approach could be used to improve both the relevance and theoretical development in tourism and hospitality research in the area of sustainability. The paper argues that conducting a systems analysis of the topic or problem of interest to the researcher will direct them to questions that offer greater relevance to both practice and theoretical development. This does not mean that the methodology of specific studies should be designed based on some sort of systems thinking process, but rather than a better understanding of the phenomenon of interest through systems thinking will result in better research questions that guide better methodological decisions. The paper will also argue that systems thinking can be applied at two levels. The first is developing an understanding of the phenomenon under study so that research can focus on leverage points that make a difference to practice. The second is developing an understanding of the theoretical systems that surround the concepts being used in the research. This paper will selectively review recent published research into tourism's social impacts to demonstrate the power of taking a systems approach at the first level. It will then critically review the use of concepts from psychology in published research into guest support for, or compliance with, sustainability programs in hospitality businesses to demonstrate the value of systems thinking at the second level.

## 2. Systems Thinking

McCool (2019) argues that the relationship between tourism and sustainability is a complex and wicked problem that can only be properly analysed using systems thinking [15]. McCool (2019) describes systems thinking as a framework that focusses on the whole seeking to identify patterns of change based on dynamic interrelationships [15]. McCool (2019) further defines systems thinking by listing its key characteristics or features and contrasting these with traditional approaches in tourism and hospitality research (see Table 1) [15]. Systems thinking is increasingly evident in research across a number of areas including sustainability science [16–18], education [19], and business management [20]. There is a small but growing body of literature using systems thinking in tourism based around work by Baggio and colleagues [21–23] looking at mapping out complex tourism systems and McCool and colleagues [15,24,25] researching tourism impacts on protected area management. This work has focused almost exclusively on developing systems mod-

els of tourism and has not been adopted more generally across other aspects of tourism research and/or used to guide the development of research questions.

**Table 1.** Features of Systems Thinking Contrasted with Traditional Research Approaches.

| Systems Thinking Features | Traditional Research Approaches |
| --- | --- |
| A focus on understanding the whole first as the key properties that emerge from the functioning of the whole system cannot be predicted from an analysis of its parts | A focus on understanding parts and assuming these build to a whole and therefore include key properties |
| A focus on the connectedness of actors and their actions | A focus on identifying, classifying and describing actors and their actions |
| Assumes nonlinear causality that contributes to continuous change through feedback loops | Assumes unidirectional causal connections between a single or small set of causes linked to a predicted or known effect |
| Is driven by a desire to change the system and emergent properties in some way | Is driven by a desire to describe the system |
| Simplifies complex systems using relatively simple models bounded by a specific problem or desired outcome | Builds increasingly complex models guided by a desire to describe in detail the processes considered to be of interest |

Sources: [15,20,24–31].

Of particular importance to the present discussion is the importance of understanding and then reducing complex systems to relatively simple systems models that focus on the key properties and outcomes from the system [15,31]. Systems models typically include:

- Causal relationships expressed as interconnections between elements:
- Elements or components which correspond to the main institutions or actors in the system;
- Feedback loops that connect the elements and relationships to outcomes of relevance to the problem being researched and managed;
- Emergent properties which are new and often surprising elements that emerge from changes in the system;
- External forces that exert pressure on the system; and
- Recognition that systems exist within other broader systems and the level of system analysed reflects the aim of the analysis [17,18,20,26–32].

## 3. Aims and Approach of This Paper

This paper has the overall aim of addressing the problem of encouraging tourism and hospitality research to be more pragmatic thus improving both its relevance to real world problems and its theoretical value. Systems thinking is the conceptual framework for the paper. The paper will focus to on two areas relevant to the problem of tourism and hospitality and sustainability that author has been involved in—tourism's social impacts on destination resident quality of life and persuading guests to engage in and support sustainability programs being implemented in hotels and restaurants.

The paper will use systems thinking principles in a conceptual analysis of evidence gathered through two critical literature reviews. Briner and Denyer (2012) [33] describe critical reviews as those which provide an argument about the area of interest supported by a selection of key references drawn from knowledge of a larger set of publications. It is important to note that these were not systematic reviews and while the search sought to be comprehensive covering the majority of papers relevant to the argument, it was not the intention to identify every single relevant published paper. It should also be noted that the searches focused on the impacts of tourism in general on destinations, not research into the impacts of events, specific forms of tourism or specific subsets of tourists. The review of research into the social impacts of tourism began with a review of literature on tourism impacts in general with the phrase tourism impacts used in Google Scholar,

Science Direct and Proquest searches. In combination these three databases provide access to 16 tourism and hospitality journals usually listed as the main journals for research publication including Annals of Tourism Research, the Journal of Travel Research, and Tourism Management. They also offer access to relevant papers published in disciplines such as sociology and sustainability, as well as to academic book chapters. As each uses a different search approach, together they provide a more comprehensive coverage of potentially relevant material.

The first stages of this review revealed the existence of several overviews in which tourism impacts are often split into three categories, environmental, economic and a large category variously labelled social, sociocultural, and social [34–37]. This first broad search revealed that research into tourism's economic and environmental impacts was widespread [38] with social impacts generally covered in research focussed on understanding resident attitudes towards and/or perceptions of tourism. A search in the three databases listed previously using phrases including social impacts of tourism, tourism social impacts, resident attitudes towards tourism, and resident perceptions of tourism and variants of these identified several major reviews of the resident attitudes research covering the period from the 1970s through to 2017 [39–43], one review of research into the social impacts of tourism specifically [44], and a shift in focus in this area overall to examinations of tourism impacts linked to quality of life (QoL), and the related concepts of subjective wellbeing (SW), happiness and life satisfaction. This shift was evident in the publication of two reviews of QoL and wellbeing with regard to destination residents and tourism impacts [45,46]. The decision was made to rely upon the review papers available in each of these areas with additional searches for publications in the years subsequent to the coverage of the most recently published review until the end of April 2020 when the analysis reported in the present paper was conducted. Table 2 summarises the total number of papers used to support the arguments made in the present paper for both sets of searches.

**Table 2.** Summary of Numbers of Papers Reviewed.

| Topic | Review Papers | | Papers Published between Most Recent Review and End of April 2020 |
| --- | --- | --- | --- |
| | **Paper** | **No. of Papers Reviewed** | |
| Social Impacts of Tourism and Resident Attitudes | Deery et al., 2012 [44] | 41 | 16 |
| | Gursoy et al., 2019 [39] | 28 | |
| | Hadinejad et al., 2019 [40] | 90 | |
| | Harrill 2004 [41] | 55 | |
| | Nunkoo et al., 2013 [42] | 140 | |
| | Sharpley 2014 [43] | 61 | |
| QoL of Destination Communities | Hartwell et al., 2018 [45] | 40 | 21 |
| | Uysal et al., 2016 [46] | 26 | |
| Guest Engagement with Sustainability | Gao et al., 2016 [47] | 26 | 47 |
| | Moscardo 2019b [48] | 19 | |
| | Nisa et al., 2017 [32] | 9 | |

Note: Appendix A has the details of the papers published subsequent to the reviews.

The second literature review followed a similar process using the same three databases but with a much more focused set of search phrases using various combination of keywords including guest/visitor/tourist participation, engagement, compliance, with hospitality sustainability programs, strategies, actions, responsible, green, pro-environmental and ethical behaviour. Again, several review papers were identified [32,47,48]. As each of these reviews focused on a specific small subset of papers identified from their search strategies and their coverage was supplemented with a search focused on papers published between the start of 2010, as none of the reviews identified relevant papers prior to that year, and the end of April 2020. Papers identified in the database searches were filtered to focus on those that reported empirical studies of guest participation in desired actions

requested by the hospitality businesses as part of their CSR or sustainability strategies. Studies attempting to identify and profile "sustainable" market segments were excluded because there is little evidence to link the respondent characteristics used in this type of research to their actual choice behaviour [49] and because such papers typically seemed to be focused on providing evidence to support an argument that hospitality managers should adopt more sustainability programs. Moscardo (2019b) notes academics appear to be considerably behind practitioners in this regard as most hospitality businesses do not require convincing about the importance of sustainability [48].

## 4. Taking a Systems Approach to the Social Impacts of Tourism

Academic discussions of tourism impacts in general share five common characteristics:

- they are typically published in or as books providing a more descriptive rather than analytic overview of tourism impacts;
- understanding tourism impacts is seen as a critical pathway to analysing tourism and sustainability;
- tourism impacts are linked to features of tourism and tourist actions;
- mostly links between tourism and tourists and impacts are focussed at the destination level; and
- they usually use a three category classification system—economic, environmental and social impacts [34–37].

It is the third category of social impacts that is of interest to the present paper. Social impacts can be defined as "the manner in which tourism effects changes in collective and individual systems, behaviour patterns, community structures, lifestyle, and the quality of life" [36].

Based on the discussion of how tourism and tourists impact destinations and using guidance from existing systems models of tourism impacts [15,25,31,38,50,51], Figure 1 presents a preliminary systems model highlighting the elements, external pressures, causal interconnections and feedback loops that together contribute to tourism impacts on destinations. The main elements or actors in the system are in boxes and these include the Destination Marketing/Management Organisations (DMOs), Tourism Service Providers, Tourists, Residents, and Tourism Infrastructure and Experience Opportunities. The two black rounded boxes are forces on the system and these include the nature and features of global tourism such as accessibility and competition and characteristics of the destination context such as its geography, economy and political systems. The Impacts of tourism on the destination are the emergent properties of this system. Arrows indicate interconnections and if double headed are potential feedback loops. The weight of the solid lines indicates a closer or more intense connection and the dotted lines are potential but not always present interconnections. Thus in some destinations tourists and residents might have considerable contact while in others there may be very little. The basic process is a simple one, the actions of tourists and the development and operation of tourism infrastructure and services are the primary causes of impacts. These two features are, in turn, primarily influenced by tourism service providers and DMOS.

In this system decisions made by tourists, tourism service providers and DMOs are the primary leverage points for changing the impacts of tourism in the destination. Thus a change in the level and type of infrastructure offered will change the impacts both directly and indirectly through changing the nature of the tourists who visit and their actions. Changes in the behaviour of tourism service providers will have a similar effect. Changes to service providers, infrastructure and the nature of experience offered and tourist can also be generated by DMOS changing their policies, management strategies and marketing. Understanding in detail how the different actions of these actors (DMOS, tourism service providers and tourists) lead to different types and levels of impacts and how these might be changed should be the target of pragmatic research that aims to address the problem of enhancing the positive and minimising or eliminating the negative impacts of tourism. The other parts of the system are included because they must be recognised as factors that

will vary both from setting to setting and over time within settings and they may influence how these leverage points operate and how they can be changed. This means that they should be considered and either reported or included as descriptive variables in research designs. In a systems thinking approach, they should not, however, be the main focus of the research.

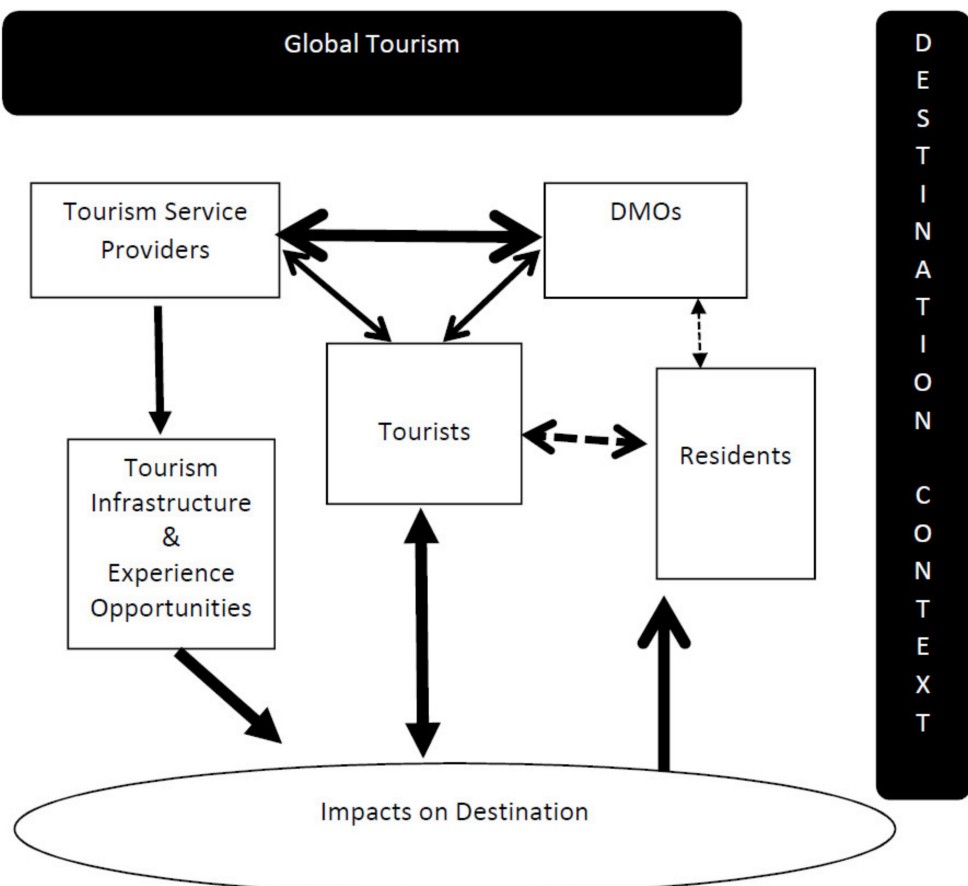

**Figure 1.** A preliminary systems model of tourism impacts on destinations.

## 5. Review of Research into the Social Impacts of Tourism

The question then becomes how much information is available on these leverage points from published academic tourism research. A search for more detail through specific empirical studies of tourism impacts in general revealed that there is considerable literature exploring environmental impacts [52] and economic impacts [53]. Two clear and consistent themes in this literature on tourism's environmental and economic impacts are that the research has focussed on the key leverage points identified in Figure 1, and the impacts studied are mostly measured objectively. The latter two themes are in stark contrast to research into tourism's social impacts which are typically measured subjectively using resident perceptions and which are focussed much more on explaining residents' perceptions of impacts using characteristics of the residents [35,36,38,54].

The review of papers on the social impacts of tourism and resident attitudes towards tourism published between 1996 and 2019 [39–44] consistently identified two major problems that continue to be evident in the 16 papers published subsequent to these reviews. Table 3 provides a summary of the key features of these 16 more recent papers. The first issue is the dominance of quantitative structured questionnaire survey methodologies [40,42,44]. According to Sharpley (2014, p. 42) [43] "*the predominantly quantitative nature of the research serves to enhance what some commentators consider to be the simplistic and theoretically weak character of much of the work on resident perceptions of tourism . . . It tends to describe what residents perceive, but does not necessarily explain why*". The second

issue is the enduring focus on explaining resident attitudes using internal characteristics of residents associated with a lack of consistent and coherent explanations for the results reported. According to Gursoy et al. (2019, p. 308) [39] "*their findings have suggested that socioeconomic factors cannot sufficiently explain the observed variations in resident attitudes toward tourism development and the identification of other latent variables is warranted*". Again, recent research into resident attitudes continues to focus on explaining resident attitudes using characteristics of residents rather than of tourism or its governance (see Table 3).

**Table 3.** Key Features of Resident Attitude Papers Published Subsequent to the Most Recent Review.

| Feature | Options | No. of Papers (Total 16) |
| --- | --- | --- |
| Methodological Approach | Quantitative | 13 |
| | Mixed method | 2 |
| | Qualitative | 1 |
| Introductory claims about issues being addressed | Improving tourism sustainability/managing impacts | 13 |
| | Supporting tourism development | 3 |
| Systems Element Studied | Residents only | 13 |
| | Other | 3 |
| Recommendations | None | 7 |
| | Changing resident attitude through marketing campaigns | 7 |
| | Changing tourism practices | 2 |

This dominance of quantitative surveys contributes to the additional problem of assuming, typically without any overt recognition or discussion of this assumption, that resident characteristics and especially resident perceptions are adequate proxies for other variables, especially impacts, contributing to confusion over the relationship between perceived and actual impacts [38,41,54]. Several authors have noted that perceived impacts are not always the same as actual impacts [38,41,54]. For example, residents might perceive tourism as responsible for an increase in crime while an examination of reliable government statistics finds no evidence to support this perception. In some instances, it could be argued that people's perceptions are what matter most because that is what they use to guide their actions and in some instances it could be argued that the only way to measure some aspects of social impacts is to ask individuals. The problem with the current research in this area in tourism is that in 13 of the 16 papers, respondent perceptions are used for nearly everything including many variables that could be measured objectively, such as the stage of tourism development, and many variables that residents, especially those with limited direct experience of tourism, cannot reliably report on.

The dominance of quantitative surveys and the use of resident perceptions as the main measures for everything studied, regardless of validity, both contribute to and result from an almost exclusive focus on explaining resident perceptions using the internal characteristics of the residents rather than the external characteristics of the tourism [39,41–44]. This continues to be an issue in recent research into resident attitudes towards tourism with 12 of the 16 papers analysing resident characteristics such as gender, community attachment, religiosity, and political views as key independent variables to explain resident attitudes towards tourism. Seven also included resident perceptions as proxies for other external characteristics with items such as perceived personal economic benefit as a proxy for actual economic benefits and perceived nature of interactions with tourists rather than actual interactions with tourists.

The most common and confused use of such a proxy is the variable of distance of residence from the main tourism precincts which was included in several papers. As noted by Sharpley (2014) [43] results linked to this measure are often contradictory and confusing, the fourth problem listed previously. In large part this is because these proxy measures are linked to different mechanisms by which tourism is assumed to connect

to residents. Therefore, some authors argued that closer residence exposes residents to more of the negative impacts of tourism such as crowding, while others argued that closer residence is linked to a greater likelihood of working in tourism and through that to greater awareness of positive impacts. In three cases in the review the reported results were contradictory to those predicted, but the authors either offered the exact opposite explanation for these contradictory results or explained them away by saying that different destinations have different geographies. In other words because there is no clear argument as to why distance between residence and tourist centres matters, any results can be explained and context moderators can be called upon in an ad hoc fashion when required. These problems exist because of the inappropriate use of resident perceptions as proxy measures for other variables and by the treatment of context moderating variables as core independent variables.

Combining the conclusions of the review papers and the examination of the papers published subsequent to these reviews indicates that nearly all the papers have examined in detail only one element (residents) in the system set out in Figure 1 and sought to explain these with a combination of features of the residents themselves and the destination context. Only a few studies have actually examined the other elements of the system or the critical leverage points. Consequently, the bulk of the research into resident attitudes towards tourism contributes nothing to our understanding of how tourism has impacts on the social dimensions of a destination. One response to this criticism is that it is not the intention of these researchers to develop an understanding of tourism impacts but rather their focus is entirely and solely on resident attitudes. All of the papers, however, were introduced with an argument that tourism impacts are important for tourism developers and planners to understand and all of them claimed to have applied implications specifically for tourism developers, planners, managers and policymakers. Seven made a claim of tourism planning and management implications but then did not actually specific what these implications were. Seven made such a claim but then argued that the implications were that DMOs should engage in education campaigns to change resident attitudes by highlighting the positive benefits of tourism. Only two papers reported on issues that the majority of destination residents linked to tourism and made suggestions on how DMOs could address these with one of these actually asking for resident support for different tourism management options [55].

A group of three papers provide some exceptions to these problems. These papers were focussed on examining actual tourist-resident interactions, included qualitative research methods, and explored how the nature of tourists and their behaviours, and the actions of DMOs and tourism businesses influenced these interactions [56–58]. In short, these papers studied at least one key leverage point in the systems model in Figure 1 and offered details on the processes that link aspects of tourism to impacts on the destination.

Sharpley (2014) [43] proposed that moves towards examining the links between tourism and residents' perceptions of their QoL and/or related concepts offers a possible way to move towards understanding the links between features of tourism and tourists and social impacts on destination residents. This is an argument also proposed by some of the earlier papers that explored tourism impacts on destination communities [54,59]. The literature search identified 2 major reviews [45,46] and 21 additional papers in journals and edited books on QoL and related concepts linked to tourism impacts on destination communities. Table 4 provides a summary of the key features of these papers and the full list can be found in Appendix A. Hartwell and colleagues (2018) [45] reported that many papers either replaced overall attitudes towards tourism with a measure of QoL/wellbeing or added some measure of QoL/wellbeing as a mediating variable but still focussed on explaining resident perceptions using characteristics of the residents alone and this was the case in eight of the additional 21 papers reviewed. The same problems also existed in terms of dominance of quantitative methods and reliance on subjective measures alone with 19 out of the 21 recent papers taking a quantitative approach with 10 relying solely on subjective measures and the majority of papers in the two reviews sharing these two

features. There was, however, movement towards understanding the leverage points identified in the Figure 1 systems models with more than half attempting to map out interconnections between the characteristics of tourism and tourists in the destinations to changes in, usually measured with both objective and subjective indicators, and various aspects of QoL/Wellbeing. Additionally three papers [38,60,61] were guided by systems models providing them with the opportunity to provide more detailed and specific recommendations to practitioners on how to change the processes of tourism to improve outcomes for destination residents.

**Table 4.** Key Features of Destination Community QoL Papers Published Subsequent to the Most Recent Review.

| Feature | Options | No. of Papers (Total 21) |
|---|---|---|
| Methodological Approach | Quantitative Only | 19 |
| | Mixed Method/Qualitative | 2 |
| Systems Element Studied | Residents Only | 8 |
| | Other | 13 |

## 6. Taking a Systems Approach to Psychology for Guest Engagement with Sustainability in Hotels and Restaurants

Kuhn's (1962) [62] much cited text on the nature of disciplines and paradigms argued that each discipline is defined by a set of core theories and accepted epistemology organised into a specific system or structure, a paradigm, which addresses recognised problems in a field. Thus, it could argued that a discipline is built upon a system made up of shared problems and accepted theories. According to Smith (2018, p. 3) [63] understanding this system of knowledge "is important for those engaged in learning the theories of the discipline and for those developing knowledge expanding the discipline". In the case of understanding hospitality guest engagement with sustainability programs the discipline is psychology and thus tourism and hospitality researchers need to understand not just the single concept that they have selected from psychology but how it fits into the broader theoretical systems that psychology researchers share. This particular applied problem of guest engagement could also be studied within the discipline of sociology but preliminary reading of the published literature shows that researchers have chosen psychology through either their focus on individual agency, a core characteristic that distinguishes psychology from sociology [64], and/or through their choice of psychological concepts and theories to apply in their empirical work.

Psychologists recognize that there is a close link between systems and theories [65–67]. Kerlinger and Lee (2000, p. 11) [68] define a theory as "*a set of interrelated constructs (concepts), definitions, and propositions that present a systematic view of phenomena by specifying relations among variables, with the purpose of explaining and predicting the phenomena*". If we replace constructs/concepts with elements, propositions with feedback loops and relations among variables with interrelationships between elements, the definitions of a theory and of a systems model are very similar. Both also share the goal of understanding this structure so as to predict and change outcomes. It could be argued that the core difference between systems and theories is that systems are more closely connected to real world problems or processes and thus are broader in their scope with theories explaining the specific elements of systems [67].

Myers and Dewall (2017) [69] describe psychology as built around two core content ideas—dual processing and taking a biopsychosocial approach to understanding human behaviour. Dual processing is a fundamental concept across all branches of psychology and argues that humans have two modes of cognition—a deep, systematic, mindful, type 1, slow processing mode which can be contrasted with a shallow, heuristic, mindless, type 2, fast processing mode [69,70]. Much of the time, and especially for habitual and routine behaviours, we engage in fast heuristic shallow processing with little or no attention paid to our actions and we reserve our cognitive processing capacity for more deliberative actions

such as complex or higher cost decisions [71]. A biopsychosocial approach recognises that our behaviours are influenced by our biology, our cognition or thoughts and emotions, and the sociocultural context we are in. Humans construct their social reality based on their personality, values, experiences, beliefs and attitudes and these social realities are shaped by relationships, culture and the physical context and surroundings [72]. These core ideas establish two critical baseline conditions of relevance to the present discussion. The first is that theories and concepts used to explain habits are fundamentally different to those used to explain deliberative actions. The second is that multiple variables impact on all actions and to expect a single or limited set of variables to make a significant difference is inappropriate [73]. This understanding of core concepts in psychology suggests that we need two different more specific systems models—one for deliberative decisions and actions and one for habits. Figure 2 provides an outline of the system model for deliberative actions, while Figure 3 provides the systems model for habitual action. The elements in these two systems models were developed using reviews of attitude intention behaviour research in general and specifically for pro-social, pro-environmental and responsible and sustainable consumption all of which take a biopsychosocial approach [74–78].

As in the previous systems model the main elements are in boxes, lines indicate interconnections and the weight of the line indicates the intensity or importance of the connection. Table 5 lists the core features of each of the elements in the first column. The critical leverage points in this system are the effectiveness of persuasive communication at changing awareness, acceptance and attitudes, the strong links between ability and action, physical setting and organisation and action, personal characteristics and social context and ability, and between action and those personal characteristics. This last leverage point recognises that engaging in an action can change the whole rest of the system by changing self-identity, self-efficacy, social reference groups and awareness and knowledge. Changing the system in practice means changing these leverage points and so pragmatic research should be focussed on these.

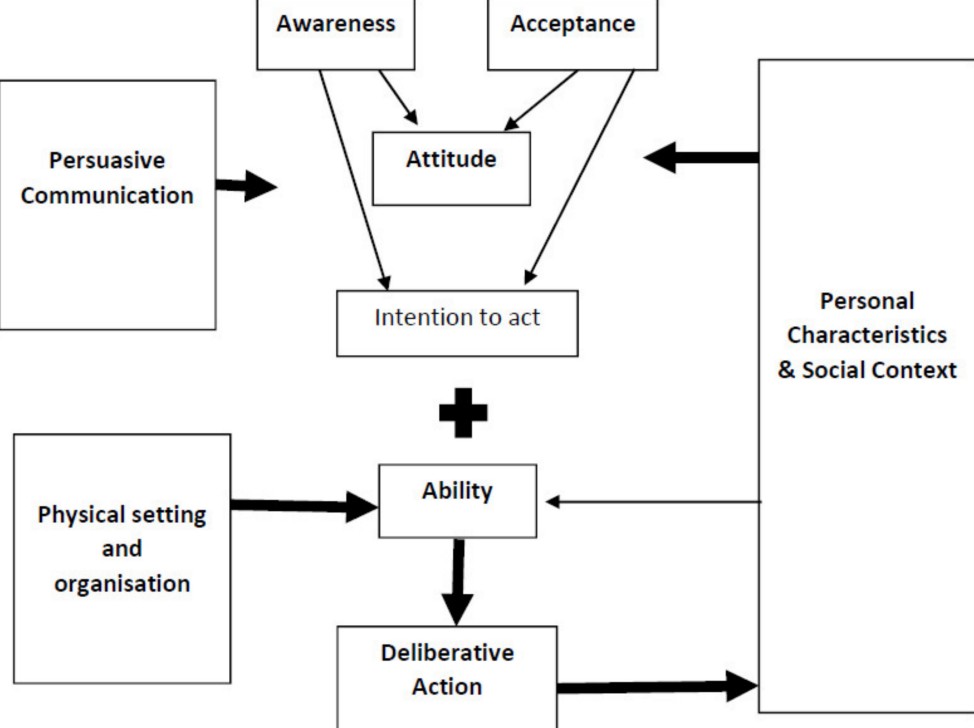

**Figure 2.** Systems Model for Deliberative Action.

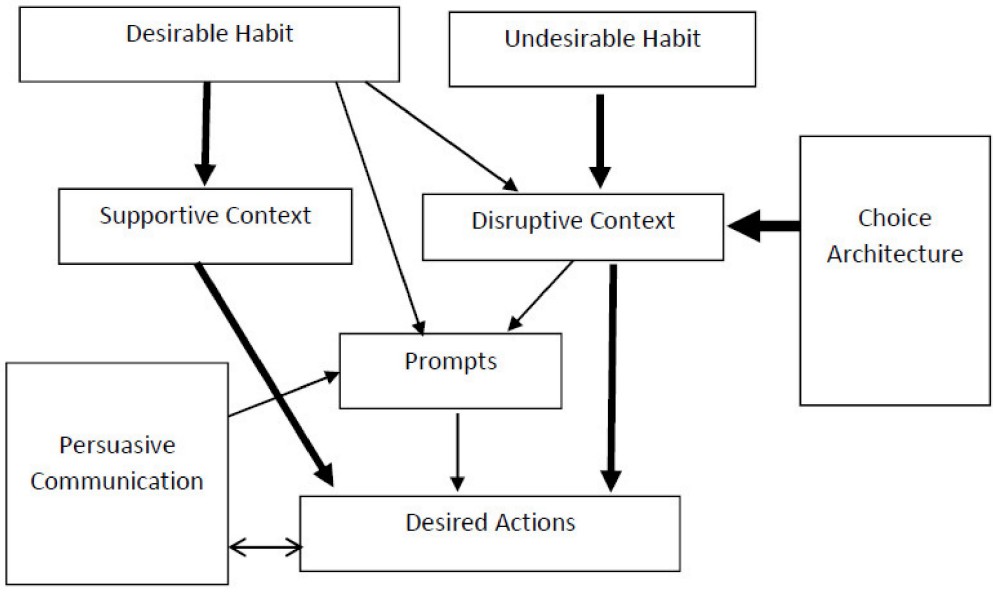

**Figure 3.** Systems Model for Habitual Action.

**Table 5.** Features of Elements and Theories Suggested in Psychology to Explain These.

| Element | Features | Main Theories |
|---|---|---|
| Personal characteristics and social context | Personality<br>Interest/motives<br>Self-efficacy<br>Values<br>Social acceptability<br>Previous experiences | Personality theory<br>Social identity theory<br>Value theory |
| Awareness | Knowledge of issue<br>Knowledge of desirable actions | See persuasive communication |
| Acceptance | Belief actions will make a difference<br>Trust in source of information<br>Acceptance of responsibility<br>Perceived social desirability or acceptability of action | Norm activation theory<br>Trust |
| Attitude | Importance and accessibility of attitude | Attitude Theory |
| Intention | Intention | Theory of Planned Behaviour (TPB) |
| Ability | Perceived behavioural control over the action<br>Self-efficacy<br>Resources—time, money and physical infrastructure | Choice architecture |
| Persuasive communication | Communication source credibility, trustworthiness and likeability<br>Communication medium accessibility and usage<br>Type of information included<br>Nature of argument presented<br>Ease of comprehension | Elaboration Likelihood Model<br>Mindfulness theory<br>Prospect theory (framing)<br>Construal theory (fluency) |
| Physical setting and context organisation | Infrastructure provided<br>Administrative procedures | Choice Architecture |

Table 5 also lists the main theories and concepts that have been identified in the psychology reviews previously listed. As can be seen there are very many theories and concepts listed and the list is not a particularly comprehensive one. Taking a biopsychosocial approach means that researchers cannot simply pick one concept or theory and expect it to explain a major dependent variable or element on its own. This does not mean that researchers must measure every possible variable in any given study, but it does mean that

core related variables need to be at least acknowledged and controlled for. Psychologists spend considerable effort to identify these related variables and remove them through experimental design or statistical control.

Figure 3 provides the system model for habits or routine actions. As can be seen it is a much simpler system. Where the established habit is the desirable action, and it is not disrupted by the physical setting then literally no action is required and research is unnecessary. Where the established habit is desirable and is disrupted then the solution lies in removing the disruption from the setting and again no research is required. Where the established habit is undesirable then choice architecture decisions can be used to make the undesirable action difficult to engage in. The model includes persuasive communication for two reasons. The first is to encourage the desired action amongst those guests who may be triggered by other factors to be mindful about their habits, although this is unlikely to be common. The second is to act as a method of rewarding desirable action and encouragement after the action to change habits. While this is an interesting element of the system it is not a key or critical leverage point. The critical leverage point in this system is the context and, as in the deliberative action system, this is most directly explained by choice architecture. Therefore, in this system there are only three critical types of research required—identifying disruptions to desirable habits, comparative evaluations of the effectiveness of different choice architecture decisions to remove these disruptions or to discourage undesirable habits, and evaluations of the effectiveness of prompts in reminding people of the action. In terms of theories relevant to these research questions, the only relevant ones are these linked to salience of prompts and choice architecture.

## 7. Review of Research Using Psychological Concepts to Examine Guest Engagement with Sustainability in Hotels

Using the systems models described in the previous section, the 47 papers on empirical studies of guest engagement in, or compliance with, sustainability strategies were divided into two groups—those that had a deliberative action (decisions about selecting and paying more for a sustainable/green hotel or restaurant, making a donation to a charity endorsed by the hotel or restaurant, deciding to reduce their personal waste and recycling) as their main research focus (30 of the 47 papers) and those that had an habitual or routine behaviour (towel reuse, switching off lights, turning off taps, shortening showers) as their main research focus (17 of the 47 papers). Recycling was included as a deliberative action even though for many people it could be seen as routine behaviour with recycling made mandatory in their workplaces and homes. This was done in part to recognize that recycling practices are not universal and in part to give researchers in this area the benefit of the doubt in judging their research decisions.

While it is not the aim of the present paper to offer more detailed methodological critiques of these 47 papers overall, it worth noting two major issues that are relevant to mapping the state of the research against the systems models presented in Figures 2 and 3. Firstly, ten studies examined intention to engage in habits which is illogical as habits are predicted primarily from existing routines, a finding noted by those four studies that included some element of habit in their paper. Secondly, across all the papers was that only three used recent psychology references when referring to psychological theories or concepts. The overwhelming majority of papers mentioned a concept and used a single psychology reference, typically 30 to 50 years out of date. Nor surprisingly few provided reasonable definitions of the concept and this often meant it was poorly understood, misapplied and inappropriately measured.

For those researchers who focussed on deliberative behaviours we see that the bulk of the research provides information for only one step in the process-intentions. Like the social impacts research all of the papers claim to have implications for management practice and/or for improving the sustainability of hotels and restaurants. Given the bulk of the research is focussed on studying only one component of the systems model for deliberative action and only a very few focussed their attention on the other critical leverage points in the systems, most of this research lacks relevance.

Finally, Table 6 repeats the list of major elements and theories used to explain deliberative action with those theories that have been used in the reviewed papers highlighted in different font styles. Those theories used in less than six papers are underlined, those used in 6-10 papers are in bold, all the others have not been considered in the reviewed research. Overall, the reviewed research has covered only a very limited number of variables. With the exception of studies using the TPB, most research seems to have selected a psychology concept at random and combined it in a haphazard fashion to another concept and sought to explain a dependent variable that is only tangentially linked to the system outcomes. Many of the quasi-experimental scenarios used in these types of studies described how levels of the core concepts varied but not how the other equally likely to matter variables that existed within these scenarios were controlled for. Poor choice of psychology concepts and a focus on a single or small set of variables without any attempt to control for others further undermines both the practical and theoretical relevance of the research.

**Table 6.** Coverage of Theories in Guest Engagement Research.

| Element | Features | Main Theories |
|---|---|---|
| Personal characteristics and social context | Personality <br> Interest/motives <br> Self-efficacy <br> **Values** <br> Social acceptability <br> Previous experiences | Personality theory <br> Social identity theory <br> **Value theory** |
| Awareness | Knowledge of issue <br> Knowledge of desirable actions | See persuasive communication |
| Acceptance | Belief actions will make a difference <br> Trust in source of information <br> Acceptance of responsibility <br> Perceived social desirability or acceptability of action | **Norms** <br> Trust |
| Attitude | Importance and accessibility of attitude | Attitude Theory |
| **Intention** | **Intention** | **Theory of Planned Behaviour (TPB)** |
| Ability | **Perceived behavioural control** <br> Self-efficacy <br> Resources—time, money and physical infrastructure | Choice architecture |
| Persuasive communication | Communication source credibility, trustworthiness and likeability <br> Communication medium accessibility and usage <br> Type of information included <br> Nature of argument presented <br> Ease of comprehension | Elaboration Likelihood Model <br> Mindfulness theory <br> Prospect theory (framing) <br> Construal theory (fluency) |
| Physical setting and context organisation | Infrastructure provided <br> Administrative procedures | Choice Architecture |

Note: Concepts underlined were used in 1–6 papers, concepts in bold were used in 6–10 papers.

## 8. Conclusions and Implications for Improving Tourism Research

In both of the reviewed areas, tourism's social impacts on destinations and guest engagement in hospitality sustainability programs, the analysis suggested that there were issues with a lack of practical relevance and inappropriate or deficient theoretical understanding. In summary, despite several hundred studies aimed at understanding tourism's social impacts on destination communities conducted over approximately 40 years, tourism academics have not generated much understanding of the processes that result in tourism

impacts. Some recent studies into resident attitudes and tourism impacts on residents QoL, however, have focussed their attention on key leverage points in the relevant system. Although limited in number, these few studies have indicated that both fast paced and intensive tourism growth generates more negative than positive impacts, that negative interactions between residents and tourists generates more negative impacts, and that characteristics of the tourists are more powerful explanatory variables than characteristics of the residents. Further the evidence taken as a whole supports the core arguments made in the Figure 1 systems model that the key leverage points are DMOs including tourism policymakers and planners, tourism service providers, including tourism developers and tourists and that characteristics of the destination itself and its residents are, at best, moderating variables.

Research into guest engagement with hospitality sustainability programs is much less extensive and the work is much more recent. It does however exhibit many of the same issues identified for the research into tourism's social impacts. There has been only a handful of studies that have focussed on key leverage points so the nearly 50 available studies have collectively provided very little new knowledge of value. This area is further undermined by some serious methodological issues. Like the first area there is an overreliance on a single methodology, poor measures of critical variables and confusion over the nature and explanatory power of the variables selected. In this second case the systems analysis of the psychological theories relevant to the research topics revealed a very poor grasp of psychology fundamentals amongst hospitality researchers leading to severely compromised methods. The few studies that avoided these issues confirmed the key elements of the two systems models with the two studies of choice architecture indicating that changing the physical context can improve guest engagement. In addition, the studies using the TPB to explain intention choose a sustainable hotel or restaurant reporting generally good explanatory power using this theory. Both Gao et al. (2016) [9] and Moscardo (2019b) [48] note, however, that these findings with regard to the TPB merely confirm what is already well-established elsewhere raising the question of its usefulness in general.

How can researchers change to move more towards being practical? This paper argues that the answer lies in developing and using systems thinking skills to better map out the phenomenon under study before selecting research questions and methodologies. The mapping out of the system of interest should alert researchers to both more valuable questions but also to the larger context. In both reviewed areas a greater focus on the key leverage points for research would significantly improve the relevance of the research. A common theme in discussions of research practice is the need for more interaction between researchers and practitioners, with practitioners more closely involved in the research and greater academic attention paid to evaluations of practice and the conduct of action research [79,80].

The examination of psychology as a set of theories also suggests that tourism and hospitality researchers need to be more careful in their use of concepts and theories from disciplines they are not trained in. More extensive reading of more contemporary references is critical. Again, taking a systems approach is also important to help guide that reading and the choice of theories and concepts to use.

**Funding:** This research received no external funding.

**Data Availability Statement:** The data presented in this study are the papers reviewed and these are available in Appendix A.

**Conflicts of Interest:** The author declares no conflict of interest.

## Appendix A. References Examined for the Critical Review Components

*Appendix A.1. References January 2018 to April 2020 on Resident Attitudes Towards/Perceptions of Tourism and Its Impacts (Not Including QoL or Similar Concepts)*

1. Alrwajfah, M.M.; Almeida-García, F.; Cortés-Macías, R. Residents' perceptions and satisfaction toward tourism development: A case study of Petra Region, Jordan. *Sustainability* **2019**, *11*, 1907.
2. Cardoso, C.; Silva, M. Residents' perceptions and attitudes towards future tourism development. *Worldw. Hosp. Tour. Themes* **2018**, doi:10.1108/WHATT-07-2018-0048.
3. Eusébio, C.; Vieira, A.L.; Lima, S. Place attachment, host–tourist interactions, and residents' attitudes towards tourism development: The case of Boa Vista Island in Cape Verde. *J. Sustain. Tour.* **2018**, *26*, 890–909.
4. Gannon, M.; Rasoolimanesh, S.M.; Taheri, B. Assessing the mediating role of residents' perceptions toward tourism development. *J. Travel Res.* **2020**, doi:10.1177/0047287519890926.
5. Litvin, S.W.; Smith, W.W.; McEwen, W.R. Not in my backyard: Personal politics and resident attitudes toward tourism. *J. Travel Res.* **2020**, *59*, 674–685.
6. Liu, X.R.; Li, J.J. Host perceptions of tourism impact and stage of destination development in a developing country. *Sustainability* **2018**, *10*, doi:10.3390/su10072300.
7. Martín, H.S.; De los Salmones Sanchez, M.M.G.; Herrero, Á. Residents' attitudes and behavioural support for tourism in host communities. *J. Travel Tour. Mark.* **2018**, *35*, 231–243.
8. Peters, M.; Chan, C.S.; Legerer, A. Local perception of impact-attitudes-actions towards tourism development in the Urlaubsregion Murtal in Austria. *Sustainability* **2018**, *10*, doi:10.3390/su10072360.
9. Rasoolimanesh, S.M.; Ringle, C.M.; Jaafar, M.; Ramayah, T. Urban vs. rural destinations: Residents' perceptions, community participation and support for tourism development. *Tour. Manag.* **2017**, *60*, 147–158.
10. Rua, S.V. Perceptions of tourism: A study of residents' attitudes towards tourism in the city of Girona. *J. Tour. Anal.* **2020**, *27*, 165–184.
11. Shtudiner, Z.E.; Klein, G.; Kantor, J. How religiosity affects the attitudes of communities towards tourism in a sacred city: The case of Jerusalem. *Tour. Manag.* **2018**, *69*, 167–179.
12. Thyne, M.; Watkins, L.; Yoshida, M. Resident perceptions of tourism: The role of social distance. *Int. J. Tour. Res.* **2018**, *20*, 256–266.
13. Tsaur, S.H.; Yen, C.H.; Teng, H.Y. Tourist–resident conflict: A scale development and empirical study. *J. Destin. Mark. Manag.* **2018**, *10*, 152–163.
14. Wassler, P.; Nguyen, T.H.H.; Schuckert, M. Social representations and resident attitudes: A multiple-mixed-method approach. *Ann. Tour. Res.* **2019**, *78*, 102740.
15. Yang, J.; Ryan, C.; Zhang, L. Social conflict in communities impacted by tourism. *Tour. Manag.* **2013**, *35*, 82–93.
16. Zamani-Farahani, H.; Musa, G. The relationship between Islamic religiosity and residents' perceptions of socio-cultural impacts of tourism in Iran: Case studies of Sare'in and Masooleh. *Tour. Manag.* **2012**, *33*, 802–814.

*Appendix A.2. References 2016–2020 on Tourism Impacts on Resident QoL/Wellbeing and Related Concepts*

1. Bimonte, S.; Faralla, V. Does residents' perceived life satisfaction vary with tourist season? A two-step survey in a Mediterranean destination. *Tour. Manag.* **2016**, *55*, 199–208.
2. Croes, R.; Ridderstaat, J.; Van Niekerk, M. Connecting quality of life, tourism specialization, and economic growth in small island destinations: The case of Malta. *Tour. Manag.* **2018**, *65*, 212–223.
3. Eslami, S.; Khalifah, Z.; Mardani, A.; Streimikiene, D.; Han, H. Community attachment, tourism impacts, quality of life and residents' support for sustainable tourism development. *J. Travel Tour. Mark.* **2019**, *36*, 1061–1079.
4. Eusebio, C.; Carneiro, M. Impact of tourism on residents' quality of life. In *Best Practices in Hospitality and Tourism Marketing and Management*; Campón-Cerro, A.M., Hernández-Mogollón, J.M., Folgado-Fernández, J.A., Eds.; Springer: Cham, Switzerland, 2019; pp. 133–158.
5. Hanafiah, M.H.; Azman, I.; Jamaluddin, M.R.; Aminuddin, N. Responsible tourism practices and quality of life: Perspective of Langkawi Island communities. *Procedia Soc. Behav. Sci.* **2016**, *222*, 406–413.
6. Konovalov, E.; Murphy, L.; Moscardo, G. An Exploration of Links between Levels of Tourism Development and Impacts on the Social Facet of Residents' Quality of Life. In *Best Practices in Hospitality and Tourism Marketing and Management*; Campón-Cerro, A.M., Hernández-Mogollón, J.M., Folgado-Fernández, J.A., Eds.; Springer: Cham, Switzerland, 2019; pp. 77–107.
7. Li, R.; Peng, L.; Deng, W. Resident Perceptions toward Tourism Development at a Large Scale. *Sustainability* **2019**, *11*, doi:10.3390/su11185074.

8.  Liang, Z.X.; Hui, T.K. Residents' quality of life and attitudes toward tourism development in China. *Tour. Manag.* **2016**, *57*, 56–67.
9.  Mathew, P.V.; Sreejesh, S. Impact of responsible tourism on destination sustainability and quality of life of community in tourism destinations. *J. Hosp. Tour. Manag.* **2017**, *31*, 83–89.
10. Moscardo, G.; Konovalov, E.; Murphy, L.; McGehee, N.G.; Schurmann, A. Linking tourism to social capital in destination communities. *J. Destin. Mark. Manag.* **2017**, *6*, 286–295.
11. Naidoo, P.; Sharpley, R. Local perceptions of the relative contributions of enclave tourism and agritourism to community well-being: The case of Mauritius. *J. Destin. Mark. Manag.* **2016**, *5*, 16–25.
12. Porras-Bueno, N.; Plaza-Meijia, M.; Vargas-Sanchez, A. Quality of life and perceptions of the effects of tourism. In *Best Practices in Hospitality and Tourism Marketing and Management*; Campón-Cerro, A.M., Hernández-Mogollón, J.M., Folgado-Fernández, J.A., Eds.; Springer: Cham, Switzerland, 2019; pp. 109–132.
13. Pratt, S.; McCabe, S.; Movono, A. Gross happiness of a 'tourism' village in Fiji. *J. Destin. Mark. Manag.* **2016**, *5*, 26–35.
14. Ramkissoon, H.; Mavondo, F.; Uysal, M. Social involvement and park citizenship as moderators for quality-of-life in a national park. *J. Sustain. Tour.* **2018**, *26*, 341–361.
15. Ridderstaat, J.; Croes, R.; Nijkamp, P. The tourism development–quality of life nexus in a small island destination. *J. Travel Res.* **2016**, *55*, 79–94.
16. Rivera, M.; Croes, R.; Lee, S.H. Tourism development and happiness: A residents' perspective. *J. Destin. Mark. Manag.* **2016**, *5*, 5–15.
17. Su, L.; Huang, S.; Huang, J. Effects of destination social responsibility and tourism impacts on residents' support for tourism and perceived quality of life. *J. Hosp. Tour. Res.* **2018**, *42*, 1039–1057.
18. Tichaawa, T.M.; Moyo, S. Urban resident perceptions of the impacts of tourism development in Zimbabwe. *Bull. Geogr. Socioecon. Ser.* **2019**, *43*, 25–44.
19. Vogt, C.; Jordan, E.; Grewe, N.; Kruger, L. Collaborative tourism planning and subjective well-being in a small island destination. *J. Destin. Mark. Manag.* **2016**, *5*, 36–43.
20. Woo, E.; Uysal, M.; Sirgy, M.J. Tourism impact and stakeholders' quality of life. *J. Hosp. Tour. Res.* **2018**, *42*, 260–286.
21. Yu, C.P.; Cole, S.T.; Chancellor, C. Resident support for tourism development in rural midwestern (USA) communities: Perceived tourism impacts and community quality of life perspective. *Sustainability* **2018**, *10*, 802.

*Appendix A.3. References Hospitality Guest Compliance with Sustainability Programs*

1.  Agag, G. Understanding the determinants of guests' behaviour to use green P2P accommodation. *Int. J. Contemp. Hosp. Manag.* **2019**, *31*, 3417–3446.
2.  Baker, M.A.; Davis, E.A.; Weaver, P.A. Eco-friendly attitudes, barriers to participation, and differences in behavior at green hotels. *Cornell Hosp. Q.* **2013**, *55*, 89–99.
3.  Balaji, M.S.; Jiang, Y.; Jha, S. Green hotel adoption: a personal choice or social pressure? *Int. J. Contemp. Hosp. Manag.* **2019**, *31*, 3287–3305.
4.  Blose, J.E.; Mack, R.W.; Pitts, R.E. The influence of message framing on hotel guests' linen-reuse intentions. *Cornell Hosp. Q.* **2015**, *56*, 145–154.
5.  Chang, H.S.; Huh, C.; Lee, M.J. Would an energy conservation nudge in hotels encourage hotel guests to conserve? *Cornell Hosp. Q.* **2015**, *57*, 172–183.
6.  Chen, H.; Bernard, S.; Rahman, I. Greenwashing in hotels: A structural model of trust and behavioral intentions. *J. Clean. Prod.* **2019**, *206*, 326–335.
7.  Chen, H.; Jai, T.M. Waste less, enjoy more: forming a messaging campaign and reducing food waste in restaurants. *J. Qual. Assur. Hosp. Tour.* **2018**, *19*, 495–520.
8.  Choi, H.; Jang, J.; Kandampully, J. Application of the extended VBN theory to understand consumers' decisions about green hotels. *Int. J. Hosp. Manag.* **2015**, *51*, 87–95.
9.  Cvelbar, L.; Grün, B.; Dolnicar, S. "To clean or not to clean?" Reducing daily routine hotel room cleaning by letting tourists answer this question for themselves. *J. Travel Res.* **2021**, *60*, 220–229.
10. Dharmesti, M.; Merrilees, B.; Winata, L. "I'm mindfully green": Examining the determinants of guest pro-environmental behaviors (PEB) in hotels. *J. Hosp. Mark. Manag.* **2020**, *29*, 830–847.
11. Dimara, E.; Manganari, E.; Skuras, D. Don't change my towels please: Factors influencing participation in towel reuse programs. *Tour. Manag.* **2017**, *59*, 425–437.
12. Dolnicar, S.; Cvelbar, L.K.; Grun, B. Do pro-environmental appeals trigger pro-environmental behavior in hotel guests? *J. Travel Res.* **2017**, *56*, 988–997.

13. Elhoushy, S. Consumers' sustainable food choices: Antecedents and motivational imbalance. *Int. J. Hosp. Manag.* **2020**, *89*, doi:10.1016/j.ijhm.2020.102554.
14. Giebelhausen, M.; Chun, H.; Cronin, J.; Hult, G. Adjusting the warm-glow thermostat: How incentivizing participation in voluntary green programs moderates their impact on service satisfaction. *J. Mark.* **2016**, *80*, 56–71.
15. Giebelhausen, M.; Lawrence, B.; Chun, H.; Hsu, L. The Warm Glow of Restaurant Checkout Charity. *Cornell Hosp. Q.* **2017**, *58*, 329–341.
16. Goldstein, N.J.; Griskevicius, V.; Cialdini, R.B. Invoking social norms: A social psychology perspective on improving hotels' linen-reuse programs. *Cornell Hosp. Q.* **2007**, *48*, 145–150.
17. Grazzini, L.; Rodrigo, P.; Aiello, G.; Viglia, G. Loss or gain? The role of message framing in hotel guests' recycling behaviour. *J. Sustain. Tour.* **2018**, *26*, 1944–1966.
18. Han, H. Travelers' pro-environmental behavior in a green lodging context: Converging value-belief-norm theory and the theory of planned behavior. *Tour. Manag.* **2015**, *47*, 164–177.
19. Han, H. Theory of green purchase behavior (TGPB): A new theory for sustainable consumption of green hotel and green restaurant products. *Bus. Strategy Environ.* **2020**, *29*, 2815–2828.
20. Han, H.; Chen, C.; Lho, L.H.; Kim, H.; Yu, J. Green Hotels: Exploring the Drivers of Customer Approach Behaviors for Green Consumption. *Sustainability* **2020**, *12*, doi:10.3390/su12219144.
21. Han, H.; Chua, B.L.; Hyun, S.S. Eliciting customers' waste reduction and water saving behaviors at a hotel. *Int. J. Hosp. Manag.* **2020**, *87*, doi:10.1016/j.ijhm.2019.102386.
22. Han, H.; Hyun, S.S. What influences water conservation and towel reuse practices of hotel guests? *Tour. Manag.* **2018**, *64*, 87–97.
23. Han, H.; Moon, H.; Lee, H. Excellence in eco-friendly performance of a green hotel product and guests' proenvironmental behavior. *Soc. Behav. Personal. Int. J.* **2019**, *47*, 1–10.
24. Han, H.; Yoon, H. Hotel customers' environmentally responsible behavioral intention: Impact of key constructs on decision in green consumerism. *Int. J. Hosp. Manag.* **2015**, *45*, 22–33.
25. Hanks, L.; Zhang, L.; Line, N.; McGinley, S. When less is more: Sustainability messaging, destination type, and processing fluency. *Int. J. Hosp. Manag.* **2016**, *58*, 34–43.
26. Hwang, K.; Lee, B. Pride, mindfulness, public self-awareness, affective satisfaction, and customer citizenship behaviour among green restaurant customers. *Int. J. Hosp. Manag.* **2019**, *83*, 169–179.
27. Kallbekken, S.; Saelen, H. 'Nudging' hotel guests to reduce food waste as a win–win environmental measure. *Econ. Lett.* **2013**, *119*, 325–327.
28. Kang, K.H.; Stein, L.; Heo, C.Y.; Lee, S. Consumers' willingness to pay for green initiatives of the hotel industry. *Int. J. Hosp. Manag.* **2012**, *31*, 564–572.
29. Kim, S.B.; Kim, D.Y. The effects of message framing and source credibility on green messages in hotels. *Cornell Hosp. Q.* **2014**, *55*, 64–75.
30. Line, N.D.; Hanks, L.; Zhang, L. Sustainability communication: The effect of message construals on consumers' attitudes towards green restaurants. *Int. J. Hosp. Manag.* **2016**, *57*, 143–151.
31. Line, N.D.; Hanks, L.; Zhang, L. Birds of a feather donate together: Understanding the relationship between the social servicescape and CSR participation. *Int. J. Hosp. Manag.* **2018**, *71*, 102–110.
32. Lu, L.; Chi, C.G.Q. Examining diners' decision-making of local food purchase: The role of menu stimuli and involvement. *Int. J. Hosp. Manag.* **2018**, *69*, 113–123.
33. Mair, J.; Bergin-Seers, S. The effect of interventions on the environmental behaviour of Australian motel guests. *Tour. Hosp. Res.* **2010**, *10*, 255–268.
34. Nicolau, J.L.; Guix, M.; Hernandez-Maskivker, G.; Molenkamp, N. Millennials' willingness to pay for green restaurants. *Int. J. Hosp. Manag.* **2020**, *90*, doi:10.1016/j.ijhm.2020.102601.
35. Nimri, R.; Patiar, A.; Kensbock, S.; Jin, X. Consumers' intention to stay in green hotels in Australia: Theorization and implications. *J. Hosp. Tour. Res.* **2020**, *44*, 149–168.
36. Olya, H.G.; Bagheri, P.; Tümer, M. Decoding behavioural responses of green hotel guests. *Int. J. Contemp. Hosp. Manag.* **2019**, *31*, 2509–2525.
37. Rahman, I.; Reynolds, D. Predicting green hotel behavioral intentions using a theory of environmental commitment and sacrifice for the environment. *Int. J. Hosp. Manag.* **2016**, *52*, 107–116.
38. Shin, Y.H.; Im, J.; Jung, S.E.; Severt, K. Locally sourced restaurant: Consumers willingness to pay. *J. Foodserv. Bus. Res.* **2018**, *21*, 68–82.
39. Shin, Y.H.; Im, J.; Jung, S.E.; Severt, K. The theory of planned behavior and the norm activation model approach to consumer behavior regarding organic menus. *Int. J. Hosp. Manag.* **2018**, *69*, 21–29.

40. Tanford, S.; Kim, M.; Kim, E.J. Priming social media and framing cause-related marketing to promote sustainable hotel choice. *J. Sustain. Tour.* **2020**, *28*, 1762–1781.
41. Tang, C.M.F.; Lam, D. The role of extraversion and agreeableness traits on Gen Y's attitudes and willingness to pay for green hotels. *Int. J. Contemp. Hosp. Manag.* **2017**, *29*, 607–623.
42. Teng, C.; Chang, J. Effects of temporal distance and related strategies on enhancing customer participation intention for hotel eco-friendly programs. *Int. J. Hosp. Manag.* **2014**, *40*, 92–99.
43. Tussyadiah, I.; Miller, G. Nudged by a robot: Responses to agency and feedback. *Ann. Tour. Res.* **2019**, *78*, doi:10.1016/j.annals.2019.102752.
44. Verma, V.K.; Chandra, B.; Kumar, S. Values and ascribed responsibility to predict consumers' attitude and concern towards green hotel visit intention. *J. Bus. Res.* **2019**, *96*, 206–216.
45. Wu, L.; Gao, Y.; Mattila, A.S. The impact of fellow consumers' presence, appeal type, and action observability on consumers' donation behaviors. *Cornell Hosp. Q.* **2017**, *58*, 203–213.
46. Yadav, R.; Balaji, M.S.; Jebarajakirthy, C. How psychological and contextual factors contribute to travelers' propensity to choose green hotels? *Int. J. Hosp. Manag.* **2019**, *77*, 385–395.
47. Zhang, L. How effective are our CSR messages? The moderating role of processing flouncy and construal level. *Int. J. Hosp. Manag.* **2014**, *41*, 56–62.

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
