# Peer review of "Using Systems Thinking to Improve Tourism and Hospitality Research Quality and Relevance: A Critical Review and Conceptual Analysis"

_tourismhosp, doi:10.3390/tourhosp2010009_

Round 1
Reviewer 1 Report
This is an interesting paper. I share, with the author, the many of the views and critiques on the literature on tourism and hospitality (extensive use of survey, structural equation modelling, interviews, and lack of use and understanding of feedback, system thinking and the appropriate tools to anaylze them).
However, I have a couple of concerns. 1) the author should recognize literature that looks at tourism as a complex system (this is lacking). See for example the work of Baggio and others (just three refs, but there is plenty in looking at tourism as a complex adaptive systems, see also reference in these here).
Baggio, R. (2008). Symptoms of complexity in a tourism system. Tourism Analysis, 13(1), 1-20.
Rodolfo Baggio, Noel Scott, and Chris Cooper. "Improving tourism destination governance: a complexity science approach." Tourism Review (2010).
Baggio, J. A., & Baggio, R. (2020). Modelling and Simulations for Tourism and Hospitality: An Introduction. Channel View Publications.
The author uses scholar and pro-quest, however, these are not reliable databases when it comes to do a systematic review of the literature. For a good summary see:
Gusenbauer, M., Haddaway, N.R., 2020. Which academic search systems are suitable for systematic reviews or meta‐analyses? Evaluating retrieval qualities of Google Scholar, PubMed, and 26 other resources. Res. Synth. Methods jrsm.1378. https://doi.org/10.1002/jrsm.1378
Section 5 would benefit greatly by figures and bar charts as well as clear citations and quotes to sustain the claims made throughout the section.
Section 6 and 7 honestly, do not fit with the paper. They may be important but if that is the case, they should be merged into section 4. As standalone, they look like sections of a different paper plugged in here. Please address this.
Finally a suggestions would be to tone down the "pointless work" accusations, although they do have some merit :).
Author Response
Response to reviewer 1
Reviewer Comment: 1) the author should recognize literature that looks at tourism as a complex system (this is lacking). See for example the work of Baggio and others (just three refs, but there is plenty in looking at tourism as a complex adaptive systems, see also reference in these here).
Response: I agree that there are tourism papers that focus on complexity thinking and systems analysis but on the whole these have only been used by researchers with a specific interest in tourism systems. This paper argues that all researchers should begin with a systems approach in order to map out their research questions. This has been clarified with additional sentences in the introduction (lines 74-80). Thus a review of the existing work, while interesting, is not directly relevant to the argument and given space restriction it was not incorporated into the introduction. I agree, however, that it might seem to some readers that by default the paper was arguing that no one in tourism was interested in systems thinking and that was not my intention. Therefore I have added a short discussion and a set of references noting that system thinking work does exist in tourism (see lines 100 - 106)
Reviewer comment: The author uses scholar and pro-quest, however, these are not reliable databases when it comes to do a systematic review of the literature. For a good summary see:
Gusenbauer, M., Haddaway, N.R., 2020. Which academic search systems are suitable for systematic reviews or meta‐analyses? Evaluating retrieval qualities of Google Scholar, PubMed, and 26 other resources. Res. Synth. Methods jrsm.1378. https://doi.org/10.1002/jrsm.1378
Response: Firstly, this is not a systematic review, it is a critical review. This is stated and defined in lines 134-138. I have added a sentence making it clear that it is not a systematic review (see lines 138-139). So the comment and the reference provided are not directly relevant to the present paper.
Seconldy, the paper (line 143) list three data bases, Google Scholar, Science Direct and Proquest.
Thirdly, the reference provided by the reviewer actually supports the choice of these three databases. It lists both Science Direct and ProQuest in their conclusion when they identify the best 14 databases to use as highlighted in the following direct quote taken from Gusenbauer & Haddaway.
“These 14 can be used as principal search systems: ACM Digital Library, BASE, ClinicalTrials.gov, Cochrane Library, EbscoHost (tested for ERIC, Medline, EconLit, CINHAL Plus, SportsDiscus), OVID (tested for Embase, Embase Classic, PsychINFO), ProQuest (tested for Nursing & Allied Health Database, Public Health Database), PubMed, ScienceDirect, Scopus, TRID, Virtual Health Library, Web of Science (tested for Web of Science Core Collection, Medline), and Wiley Online Library”
They were critical of Google Scholar but that was because many of the criteria they used to judge the databases were linked to ease of use and flexibility of search rules for the user. They note that desired performance criteria need to be evaluated relative to the specific systematic search requirements of the reviewer. These criteria were not relevant to the present situation. They also note that Google Scholar uses a fundamentally different approach to searching and that combinations of databases with different approaches may be a solution to balancing out their individual weaknesses. Finally, they argue that “What records are considered relevant depends on the specific requirements of the reviewer and thus cannot be generalised. For example, a search system with a smaller size, covering only a single discipline, might bring more relevant search results than a large search system covering multiple disciplines”.
The take home message from this review of databases is that you should tailor the databases that you use to the topic areas you are interested in, combine several databases with different search approaches and that ProQuest and Science Direct are good options.
In the present study I selected the three databases using these general rules and because together they include the main 16 tourism/hospitality journals and each offers some additional searching capabilities. The table below shows that coverage. This level of detail is not appropriate for the paper itself but I have added some sentences to summarize why those three databases were chosen to the text (see lines 143-149)
Table: Linking the Databases Used to Tourism & Hospitality Journals
|
Journal |
Google Scholar |
ProQuest |
Science Direct |
|
Annals of Tourism Research |
Through direct links to Taylor & Francis Online |
|
|
|
Journal of Travel Research |
Through direct links to Sage Premier |
|
|
|
Tourism Management |
|
|
√ |
|
Tourism Management Perspectives |
|
|
√ |
|
J. Sustainable Tourism |
Through direct links to Taylor & Francis Online |
|
|
|
Cornell Quarterly |
|
|
√ |
|
Int. J. Hospitality Management |
|
|
√ |
|
J. Destination Marketing & Management |
|
|
√ |
|
J. Hospitality & Tourism Management |
|
|
√ |
|
J. Hospitality Marketing & Management |
Through direct links to Taylor & Francis Online |
|
|
|
J. of Travel & Tourism Marketing |
Through direct links to Taylor & Francis Online |
|
|
|
Current Issues in Tourism |
Through direct links to Taylor & Francis Online |
|
|
|
Tourism Recreation Research |
Through direct links to Taylor & Francis Online |
|
|
|
Tourism Review |
|
√ |
|
|
Int, J. Contemporary Hospitality Management |
|
√ |
|
|
Int. J of Tourism Research |
|
√ |
|
Notes:
- Google Scholar will also identify papers published in all of the listed journals if the authors link them in some public repository or listing as is common practice in most educational institutions
- ProQuest covers a wide range of other journals in which tourism studies are likely to be published
Reviewer Comment: Section 5 would benefit greatly by figures and bar charts as well as clear citations and quotes to sustain the claims made throughout the section.
Response: I suspect that this comment reflects the confusion over the type of review as figures and bar charts would be much more appropriate to a systematic review. Critical reviews would not normally provide that type of analysis. It is, however, a long section and so I have added a table summarizing the main points and providing the number of articles linked to each point. I have also included quotes for the review papers to support the list of main problems identified early in the section.
I did consider adding an example reference for each point but I decided not to as it would focus unfair attention on a single paper when the problem was often shared across multiple papers and there is insufficient space to provide a complete list of the relevant papers for each point. I think the inclusion of the appendices, which seem to have been missing in the paper sent out for review , also addresses the reviewer’s concerns in this comment.
Reviewer Comment: Section 6 and 7 honestly, do not fit with the paper. They may be important but if that is the case, they should be merged into section 4. As standalone, they look like sections of a different paper plugged in here. Please address this.
Response: While I think that either of the two levels discussed in the paper could be lengthened into its own paper, I believe that the argument for using a systems approach is stronger when both levels are considered. Further, the other 3 reviewers had no problem with the inclusion of both levels so I have not made any changes in response to this comment. I will leave the final decision to the editor.
Reviewer Comment: Finally a suggestions would be to tone down the "pointless work" accusations, although they do have some merit :).
Response: Removed the use of the phrase “pointless” throughout and lightened the critical tone.

Reviewer 2 Report
I read this article with great enthusiasm, as I’m a believer of system’s approaches. Furthermore, I am convinced that we do need to employ them more often to gain a more holistic understanding of certain phenomena and to identify areas for target intervention. At least from the tourism perspective. However, while reading this manuscript, I struggled with two major issues, which are interconnected, as I detail below.
- The relationship between the need (and how to) incorporate system’s approaches in tourism/hospitality research and the study methods. System’s approaches are one approach that can incorporate different stakeholders when investigating a phenomenon, as very well explained in the manuscript. However, that is not the same to say that should be the main approach, or suggesting that deviating from that norm makes a study useless. It emerged from the narrative the message that most (if not all) of the studies conducted on the social aspects of hospitality/tourism sustainability are useless. There is still need to investigate specific aspects (e.g., challenges, impacts, barriers, needs) of specific actors, and not within a system. Furthermore, I would posit… Taking into account the system’s nature of sustainability, could a system’s approach be employed to only look at the social dimension of sustainability? Some issues to consider related to this point:
- References to an Appendix A was placed in several parts of the text, but the appendix was not readily available.
- Rather than listing flaws of the papers examined (which is hard to follow), I would suggest to make a table and describe the issues encountered.
- The procedures of the systematic review were well described. But it would increase clarity to have a table summarizing how many manuscripts were found, how many were included. At the end, it seems a very few papers that were examined, which surprises me as this is a research area of wealth in the overall tourism literature.
- I found the selection of the journals used in the second literature review also troublesome for 2 reasons. (1) There were included 2 from hospitality and 1 from tourism, which itself is not proportional. (2) I can only speak for tourism, but from my opinion, there were 2 top journals (Tourism Management & J. of Travel Research) that emphasize both, theoretical and practical implications that maybe should have been included along the J. of Sustainable Tourism.
- The narrative in several places states (p. 519) “the majority of the research was pointless based on problems with methodological rigour, practical relevance and inappropriate or deficient theoretical understanding.”. However, it is not very clear what are the basis for such claim… how did the author(s) determined such lack of rigor or theoretical deficiency? A more through and organized explanation could be useful. Also, providing a table with the characteristics studied to make such a decision would be useful.
- Related to the previous point… If we considered that well recognized journals in our field (as the ones cited in the paper), usually have 2-3 rounds involving 2-3 peer reviews plus and an editor(s) with highly recognized in our fields, please can the author(s) elaborate who (how many researchers) and how (independent coders using a rubric? Follow up triangulation) perform such a review of the quality of the papers?
- Some contentions claims can also be addressed. For example, (p. 509) “ten out of the 17 papers that studied habitual behaviour had a major logical flaw in that they studied intentions to engage in these behaviours, leaving only seven with potential to offer research of value.”. It is true that we, researchers, have used intentions as proxy for actual behaviors too much… But a system’s approach would fix that? Also, it is important to recognize that for budgetary and time constraints, studying actual behaviors may require longitudinal studies that may not always be feasible.
- The tone of the narrative, rather than being constructive (i.e., how to move forward with system’s approaches) is pernicious (e.g., “Again the logical conclusion to draw is that to date very little of the published research is of any real value to either practice or theoretical development…” , p. 513). As a scholar, I do value criticism as this help us to move forward. But criticism, in my opinion, should be based on specific elements for improvement, rather than diminish existing research, especially those that have been scrutinized through peer review. In many parts of the narrative, statements can be reworded to be less unequivocal. Some examples to illustrate my point (there are many).
- This paper starts with a strong argument (based on a citation 10 years old) that can be contested. (p. 24) Tourism and hospitality, as an identified research area is not a discipline but an applied topic…”.
- This statement could be toned down and make it sound like a universal truth, like: “Tourism and hospitality, as an identified research area, can be examined from multiple disciplines”.
- Of course, an alternative is to expand on those who believe that it is a discipline.
- (lines 53-54) >> “Arguably puerile, pedantic and popularist research could be classified into a single group with the label of pointless”.
Here is another strong statement that deserves further clarification, tone down or discussion. Clarifying some definitions (pedantic research) could have important policy and managerial implications. For example, they may serve to inform policy as to protect a stakeholder (e.g., small size providers of authentic experiences from developers staging authenticity). They could also inform branding strategies to provide providers with a consistent and orchestrated message to reach their target market.
So, are these investigations pointless? Personally, I believe not necessarily. It is far beyond a published paper to examine the actual interventions or outreach efforts this research caused.
Finally, I wanted to mention that the narrative was very good. I found some minor typos in need to clean up. E.g.,
- (Line 38) >> “…light on this r gap”.
- (Line 448) >> change the comma for a period between “…action, In…”
- I found an excessive use of the term “arguably”, when actually the opposing arguments were not elaborated. However, this could be an issue of narrative style.
Author Response
Response to Reviewer 2
Reviewer Comment: The relationship between the need (and how to) incorporate system’s approaches in tourism/hospitality research and the study methods. System’s approaches are one approach that can incorporate different stakeholders when investigating a phenomenon, as very well explained in the manuscript. However, that is not the same to say that should be the main approach, or suggesting that deviating from that norm makes a study useless.
Response: It appears that my initial explanation of the argument for systems thinking is not very clearly set out. I had not intended to argue that every study must take within itself a systems thinking approach to its methodology but rather that taking a systems theory approach to the topic/problem that is being examined would improve the research questions asked and taking a systems approach to the relevant theories would improve the concepts and theories adopted. I have amended the sentences in the introduction (lines 74-75) to clarify this and explained this again the revised conclusion.
Reviewer Comment: It emerged from the narrative the message that most (if not all) of the studies conducted on the social aspects of hospitality/tourism sustainability are useless. There is still need to investigate specific aspects (e.g., challenges, impacts, barriers, needs) of specific actors, and not within a system. Furthermore, I would posit… Taking into account the system’s nature of sustainability, could a system’s approach be employed to only look at the social dimension of sustainability?
Response: I think the first part of this comment is much broader than the paper is actually arguing. The specific focus was on the social impacts of tourism on destination communities not the social aspects of sustainability more broadly. The paper did conclude in the section on that part of the review that more recent papers, particularly within the QoL/wellbeing theme, were moving towards better approaches. This was lost in an overly succinct statement at the start of the conclusion which has been amended.
The paper does not argue that investigating specific aspects of specific actors is not an appropriate research approach, but rather that studying such specifics in the absence of any acknowledgement or recognition of the larger system they operate within makes it extremely unlikely that the research will make a difference either to the practice or to the theoretical development of the field. That is the basic argument of systems theory – knowing the more about the specific elements in a system does not help us understand the system itself.
Reviewer Comment: References to an Appendix A was placed in several parts of the text, but the appendix was not readily available.
Response: I’m not sure why the appendix was not included in the document sent out for review, but it is now included in the paper itself.
Reviewer Comment: Rather than listing flaws of the papers examined (which is hard to follow), I would suggest to make a table and describe the issues encountered.
Response: 3 tables have been added as suggested.
Reviewer Comment: The procedures of the systematic review were well described. But it would increase clarity to have a table summarizing how many manuscripts were found, how many were included. At the end, it seems a very few papers that were examined, which surprises me as this is a research area of wealth in the overall tourism literature.
Response: It is important to note that this paper did not claim to have conducted a systematic literature review, it was a critical literature review. The argument also was based on the conclusions of several existing review papers supplemented by an analysis of papers published subsequent to the reviews and so the conclusions are based on larger numbers of papers than just the recent papers that were analysed. A table explaining the total numbers including the number of papers in the existing reviews used has now been added.
Reviewer Comment: I found the selection of the journals used in the second literature review also troublesome for 2 reasons. (1) There were included 2 from hospitality and 1 from tourism, which itself is not proportional. (2) I can only speak for tourism, but from my opinion, there were 2 top journals (Tourism Management & J. of Travel Research) that emphasize both, theoretical and practical implications that maybe should have been included along the J. of Sustainable Tourism.
Response: The second literature review was not based on just 3 journals – it used the three databases in the same way as for the first review but then added a check on those three journals as they were the ones where the bulk of the relevant papers were published. As that second check did not actually reveal any additional papers it has now been removed from the explanation. The addition of the appendix will also assist in clarifying this issue as it will be clear that papers from a range of journals were included. Further, an explanation of why the three databases were chosen, because they explicitly include the main 16 journals in tourism and hospitality, also contributes to clarification of this issue.
In addition I have checked Annals of Tourism Research, Tourism Management and the Journal of Travel Research individually and no additional relevant papers within the original time frames were discovered – all the relevant papers had been found using the three databases.
Reviewer Comment: The narrative in several places states (p. 519) “the majority of the research was pointless based on problems with methodological rigour, practical relevance and inappropriate or deficient theoretical understanding.”. However, it is not very clear what are the basis for such claim… how did the author(s) determined such lack of rigor or theoretical deficiency? A more through and organized explanation could be useful. Also, providing a table with the characteristics studied to make such a decision would be useful. Related to the previous point… If we considered that well recognized journals in our field (as the ones cited in the paper), usually have 2-3 rounds involving 2-3 peer reviews plus and an editor(s) with highly recognized in our fields, please can the author(s) elaborate who (how many researchers) and how (independent coders using a rubric? Follow up triangulation) perform such a review of the quality of the papers?
Response: All mentions of methodological rigour have been removed from the paper as they were distracting from the main argument. I do believe that there were in fact, multiple problems with methodology but that is a separate paper that would indeed require much more supporting evidence as noted in the comment.
Reviewer Comment: Some contentions claims can also be addressed. For example, (p. 509) “ten out of the 17 papers that studied habitual behaviour had a major logical flaw in that they studied intentions to engage in these behaviours, leaving only seven with potential to offer research of value.”. It is true that we, researchers, have used intentions as proxy for actual behaviors too much… But a system’s approach would fix that? Also, it is important to recognize that for budgetary and time constraints, studying actual behaviors may require longitudinal studies that may not always be feasible.
Response: The reviewer is confusing two separate statements, one about studying intentions in general and one about studying intentions to engage in habitual behaviour specifically. The paragraph has been revised to be clearer. There is a major logical flaw in studying intentions to engage in a habit as habitual behaviour has no intention. Intention to engage in a deliberative action is a completely different issue. It is still not a useful research approach because there is not often a very strong connection between intentions and action, especially as the behaviours get generic. But I have removed that conclusion as it is tangential to the main argument. I
However, I strongly disagree with the argument that it is okay to study one thing because it is easier than studying something else. That is like the proverbial drunk searching for his lost keys only under the street lamp at night. If the thing being studied is not connected strongly connected to the outcome of interest then studying it is indeed pointless. If the better research needs more time and money then you should either find the time and money or change area. I would also argue that in this particular case its not actually that hard to study action – as several papers did that – unfortunately many of them studied habitual action with theories for deliberative action, but they did at least study action.
Reviewer Comment: The tone of the narrative, rather than being constructive (i.e., how to move forward with system’s approaches) is pernicious (e.g., “Again the logical conclusion to draw is that to date very little of the published research is of any real value to either practice or theoretical development…” , p. 513). As a scholar, I do value criticism as this help us to move forward. But criticism, in my opinion, should be based on specific elements for improvement, rather than diminish existing research, especially those that have been scrutinized through peer review. In many parts of the narrative, statements can be reworded to be less unequivocal. Some examples to illustrate my point (there are many).
Response: It was not my intention to be pernicious, but rather to attempt to highlight what I see as significant issues that are increasing within tourism research as a whole. The tone reflects a sense of frustration with the continuing decline in the quality of published tourism research. But that is possibly not helpful, so the tone has been changed as set out in the following comments and additional material has been added to the conclusion to outline more specifically how to move forward.
Reviewer Comment: This paper starts with a strong argument (based on a citation 10 years old) that can be contested. (p. 24) Tourism and hospitality, as an identified research area is not a discipline but an applied topic…”.
- This statement could be toned down and make it sound like a universal truth, like: “Tourism and hospitality, as an identified research area, can be examined from multiple disciplines”.
- Of course, an alternative is to expand on those who believe that it is a discipline.
Response: the suggested change has been made and thus the argument about the disciplinary status of tourism has been removed. It is not central to the present paper.
Review Comment: (lines 53-54) >> “Arguably puerile, pedantic and popularist research could be classified into a single group with the label of pointless”. Here is another strong statement that deserves further clarification, tone down or discussion.
Response: The sentence has been removed and the use of the phrase pointless has been removed from the paper entirely.
Reviewer Comment: Clarifying some definitions (pedantic research) could have important policy and managerial implications. For example, they may serve to inform policy as to protect a stakeholder (e.g., small size providers of authentic experiences from developers staging authenticity). They could also inform branding strategies to provide providers with a consistent and orchestrated message to reach their target market. So, are these investigations pointless? Personally, I believe not necessarily. It is far beyond a published paper to examine the actual interventions or outreach efforts this research caused.
Response: The description of pedantic research has been changed to remove the element of definitional discussions.
Reviewer Comment: Finally, I wanted to mention that the narrative was very good. I found some minor typos in need to clean up. E.g.,
- (Line 38) >> “…light on this r gap”.
- (Line 448) >> change the comma for a period between “…action, In…”
- I found an excessive use of the term “arguably”, when actually the opposing arguments were not elaborated. However, this could be an issue of narrative style.
Response: All the suggested changes have been made and the revised paper carefully proofread. I would note though that the word arguably was only used three times - it is now not used at all.

Reviewer 3 Report
I agree with the author the “managerial perspective” has monopolized not only the knowledge production but also the foundations of epistemology in the constellations of tourism fields, without mentioning the advance of tourist-centricity –citing Franklin- which obscures more than it clarifies. To some extent, the tourist has placed as the only valid source of information leading the discipline into a gridlock. I must confess, in this context, quantitative-led methods are prioritized than the qualitative one. A third problem emerges, in the dissociation between tourism theory and practice, probably the gap this research tries to fill. The following points enumerated below aims to improve the main argumentation and readability of the manuscript.
At a close look, give an original description of the current problems of tourism research today, -in the introductory section- which justifies Anderson, Herriot & Hodginkon´s worries. I mean to a big problem with seems to be the stagnation of tourism research. The publish or perish culture undermines the quality of tourism research but if you ask me it is not limited to tourism, the same happens in sociology, anthropology or geography. Several factors that seem to be today coadjuvant in the low quality works strongly associated with the urgency of faster publications. Bob McKercher´s notion of multi-authors is part of the problem. There is no clear the connection between sustainability and the knowledge production –or in terms of author thinking system-.
For the reader, it would be best if the author summaries the strengths and weaknesses of the reviewed literature in a shortlist or description. I do not agree psychologists or epistemologists say overtly that there is a close link between theories and thinking systems. A theory should be understood as a set of discourses, expectations and assumptions which are self-explanatory for the studied problem, but probably not for other problems. The power of science to explain phenomenal events is always temporal and changes in the threshold of time. See the works of Aldred Schutz, Richard Rorty (on Pragmatism).
The notion of credibility is not defined, at the best I would dialogue with three or four authors who have focused on the philosophical nature of credibility. In consonance with the conclusion, I incline to think one of the main limitations of tourism research consists in the lack of dissociation between theory and practice. I put the example of disaster studies which fall in my scope of expertise. Disasters are disasters but the point is to define what a disaster mean?
A couple of scholars will place interest in understanding or giving an explanatory option of what a disaster means. They will deal with different theories coming from a wide range of sub-disciplines. A second academic group associates to policy-makers who are interested in drawing the most efficient measures and steps to prevent or mitigate the negative effects of disasters. In disaster studies, both groups dialogue but anyone intervenes in the fields of the other. Unfortunately, in tourism, this dissociation is far from being clear. Conceptual papers are oriented to good practices to protect tourist destination image, while tourism-management studies intend scientific rigorist. To put the same in bluntly one group looks forward to understanding the issue whereas the other works hard to come into fruition efficient instrument to resolve a manifest problem.
Author Response
Response to Reviewer 3
Reviewer Comment: I agree with the author the “managerial perspective” has monopolized not only the knowledge production but also the foundations of epistemology in the constellations of tourism fields, without mentioning the advance of tourist-centricity –citing Franklin- which obscures more than it clarifies. To some extent, the tourist has placed as the only valid source of information leading the discipline into a gridlock. I must confess, in this context, quantitative-led methods are prioritized than the qualitative one. A third problem emerges, in the dissociation between tourism theory and practice, probably the gap this research tries to fill.
Response: Thank you, I believe that this is a critical issue for tourism studies
Reviewer Comment: For the reader, it would be best if the author summaries the strengths and weaknesses of the reviewed literature in a shortlist or description.
Response: 3 tables have been added summarising the papers reviewed as suggested.
Reviewer Comment: I do not agree psychologists or epistemologists say overtly that there is a close link between theories and thinking systems.
Response: Psychologists do say overtly that there is a close link between theories and systems. The three references provided (Kern et al 202o; Marc 2010 & Sexton & Stainton 2016) are all psychologists saying that there is a link between theories and systems models of the aspects of psychology that the theories are seeking to explain. They further argue that sets of theories focussed on specific problems can be seen as types of system themselves. Psychologists routinely link models of the processes they are trying to explain with the theories that they use to explain them and there is considerable discussion about how different theories assume, implicitly or explicitly, certain models of the systems they are applied to. It is so fundamental to the way psychology researchers think that it is not often made explicit in individual study reports. But it is very clear in reviews and textbooks.
The statement is based not only on my 10 years of tertiary qualifications in psychology, publications in psychology journals and my 8 years of teaching psychology in a psychology department but also on statements made by psychologists in psychology publications. I am not disputing the definition of theories that the reviewer provides but regardless of how you define theories there are clear parallels and connections that can be made between theories and systems. The following are some additional references focussed on psychology that further support the statement.
Weiskopf, D. A. (2011). Models and mechanisms in psychological explanation. Synthese, 183(3), 313-338.
Baddeley, A. (2012). Working memory: theories, models, and controversies. Annual review of psychology, 63, 1-29.
Gentner, D., & Stevens, A. L. (Eds.). (2014). Mental models. Psychology Press.
Patton, W., & McMahon, M. (2014). Career development and systems theory: Connecting theory and practice (Vol. 2). Springer.
Schmittmann, V. D., Cramer, A. O., Waldorp, L. J., Epskamp, S., Kievit, R. A., & Borsboom, D. (2013). Deconstructing the construct: A network perspective on psychological phenomena. New ideas in psychology, 31(1), 43-53.
Mele, C., Pels, J., & Polese, F. (2010). A brief review of systems theories and their managerial applications. Service science, 2(1-2), 126-135.
Reviewer Comment: The notion of credibility is not defined, at the best I would dialogue with three or four authors who have focused on the philosophical nature of credibility.
Response: I do not understand the issue with this comment. Credibility is only mentioned once in the paper (actually source credibility - a very specific concept in its application) as one of several variables studied in persuasive communication. It is not linked in any way to the arguments in the paper and I have not provided definitions of any of the other 20 variables listed in that table. I don’t see how including a discussion of credibility definitions, and logically then definitions of the other 20 variables contributes to the paper so I’m not going to change anything in the paper. If the reviewer wants to provide further information I’m happy to consider this further as I may have misunderstood the point.
Reviewer Comment: In consonance with the conclusion, I incline to think one of the main limitations of tourism research consists in the lack of dissociation between theory and practice. I put the example of disaster studies which fall in my scope of expertise. Disasters are disasters but the point is to define what a disaster mean? A couple of scholars will place interest in understanding or giving an explanatory option of what a disaster means. They will deal with different theories coming from a wide range of sub-disciplines. A second academic group associates to policy-makers who are interested in drawing the most efficient measures and steps to prevent or mitigate the negative effects of disasters. In disaster studies, both groups dialogue but anyone intervenes in the fields of the other. Unfortunately, in tourism, this dissociation is far from being clear. Conceptual papers are oriented to good practices to protect tourist destination image, while tourism-management studies intend scientific rigorist. To put the same in bluntly one group looks forward to understanding the issue whereas the other works hard to come into fruition efficient instrument to resolve a manifest problem.
Response: I agree and I have added sentences to the conclusion making this point

Reviewer 4 Report
In general, I think, in my humble opinion, this paper is a good work,
and has an interesting theme and the research is good and correct.
Author Response
Thank you for your positive comments
Round 2
Reviewer 1 Report
I do think that the author(s) have adequately replied to my previous concerns. I am critical of critical reviews :), but this would not disqualify a paper such as this one that is well argumented.
Reviewer 2 Report
Many thanks for addressing the reviews/comments stated related to the original version. The current version reads clearly and I believe it will be of great contribution for future studies.
I especially appreciate changing the narrative style as to be less contentious. It now reads good and still provocative.